# A High-Performance MEMS Accelerometer with an Improved TGV Process of Low Cost

**DOI:** 10.3390/mi13071071

**Published:** 2022-07-05

**Authors:** Yingchun Fu, Guowei Han, Jiebin Gu, Yongmei Zhao, Jin Ning, Zhenyu Wei, Fuhua Yang, Chaowei Si

**Affiliations:** 1Beijing Smart-Chip Microelectronics Technology Co., Ltd., Beijing 100192, China; fuyingchun@icrus.cn; 2Zhongguancun Xinhaizeyou Technology Co., Ltd., Beijing 100192, China; 3Institute of Semiconductors, Chinese Academy of Sciences, Beijing 100083, China; hangw1984@semi.ac.cn (G.H.); ymzhao@semi.ac.cn (Y.Z.); ningjin@semi.ac.cn (J.N.); zywei97@semi.ac.cn (Z.W.); fhyang@semi.ac.cn (F.Y.); 4State Key Laboratory of Transducer Technology, Shanghai Institute of Microsystem and Information Technology, Chinese Academy of Sciences, Shanghai 200050, China; j.gu@mail.sim.ac.cn; 5Center of Materials Science and Optoelectronics Engineering, University of Chinese Academy of Sciences, Beijing 100049, China

**Keywords:** MEMS accelerometer, TGV, distributed stoppers, comb structure

## Abstract

High-performance MEMS accelerometers usually use a pendulum structure with a larger mass. Although the performance of the device is guaranteed, the manufacturing cost is high. This paper proposes a method of fabricating high-performance MEMS accelerometers with a TGV process, which can reduce the manufacturing cost and ensure the low-noise characteristics of the device. The TGV processing relies on laser drilling, the metal filling in the hole is based on the casting mold and CMP, and the packaging adopts the three-layer anodic bonding process. Moreover, for the first time, the casting mold process is introduced to the preparation of MEMS devices. In terms of structural design, the stopper uses distributed comb electrodes for overload displacement suppression, and the gas released by the packaging method provides excellent mechanical damping characteristics. The prepared accelerometer has an anti-overload capability of 10,000 g, the noise density is less than 0.001°/√Hz, and it has ultra-high performance in tilt measurement.

## 1. Introduction

MEMS devices have the advantage of being low cost and small in size, and they can be fabricated in batches. They are also successful sensors for various environmental monitoring uses. MEMS inclination sensors are widely used in buildings, bridges, power transmission networks, automobiles, and other engineering vehicles [1,2]. Accelerometers with high performance and high reliability have great market value.

At present, the accelerometer with the best performance is the sandwich structure from Murata and Safran. The devices of Colibris are based on three-layer silicon bonding and have excellent temperature stability [3], but the preparation cost is high, and it is often used in high-performance fields such as aerospace engineering [4]. The devices from Murata use a silicon glass composite cover plate as the upper and lower electrodes, and the electrodes are grown on the sidewalls through a hard mask, which achieves a good balance between performance and cost, and has a wide range of applications in the automotive and industrial fields [5]. However, since Murata’s structure requires electrodes to be grown on the sidewalls, it is not possible to screen the yield of devices on the wafer [6], and the cost is still higher than that of consumer-grade accelerometers.

In this paper, an accelerometer with comb-tooth structures is proposed. It is implemented based on SOG process, and the structure layer adopts 100 μm silicon to ensure the size of the mass block, thereby reducing the mechanical noise of the device. The packaged capacitor electrodes are led out to the surface of the device through vertical leads, which can meet wafer-level tests and reduce production costs. Relying on the distributed stopper set between the comb teeth, the sensitive mass displacement is constrained, depending on the distributed stopper structure set between the comb teeth. Metal filled in through vias on BF33 glasses serves as the vertical lead cover, the anodic bonding process achieves hermetic packaging, and the released gas provides additional damping, making the device excellent in vibration resistance.

The proposed process solution has the capability of low-cost mass production and can guarantee the performance of the device, which is extremely innovative.

## 2. Structure Design

From the perspective of detection accuracy, the sensitivity of capacitance change per unit acceleration needs to be considered when designing the tilt angle structure, which is determined by the resonant frequency of the structure, the distance between the capacitor plates, and the total capacitance. Under the action of inertial force, the response displacement of the accelerometer can be obtained from Newton’s second law, as follows:(1)$$x=a/\omega_{0}^{2}$$
where *x* is the displacement under the acceleration a, and *ω*_0_ is the resonant frequency of the accelerometer. If *C*_0_ is the initial capacitor, then *d*_0_ is the initial distance of capacitor plates, and the capacitance change Δ*C* is
(2)$$\Delta C=xC_{0}/d_{o}$$

The accelerometer resolution is determined by its mechanical noise, and the noise ⟨*a_n_*⟩ is related to the resonant frequency and the size of the mass *m* [7]. Where *k* is the Boltzmann constant, *T* is the ambient temperature, *m* is the mass of the movable structure, and *Q* is the Quality factor.
(3)$$\langle a_{n}\rangle =\sqrt{4kT\omega_{0}/mQ}$$

To ensure sensitivity, the typical resonant frequency of an inclination accelerometer is several hundred hertz. Considering the anti-overload capability of the accelerometer, the shock response displacement should not be too large, which is designed to be 1 KHz here. To ensure that the accelerometer has a detection accuracy of 0.001°, its output electrical noise should be less than 3 μg/√Hz, and the corresponding mechanical noise should be at the μg level.

Considering the over-damping situation, the quality factor of the accelerometer is less than 1, and the mass of the accelerometer should be greater than 0.12 mg. At present, the detection capability of the accelerometer detection circuit for differential capacitance is at the zF level. Considering the tolerance, it is hoped that 0.001° can correspond to a capacitance change rate of 10 zF, then the capacitance sensitivity of an ideal inclination accelerometer is expected to reach 0.57 pF/g.

For a 100 μm-thick silicon structure MEMS device, considering the ability of deep silicon etching, the capacitor plate spacing should be greater than 3 μm. Considering the processing accuracy, the capacitor plate spacing is set to 4 μm, the length of the plate is 150 μm, and the width is 6 μm. The overlap length is 140 μm. One anchor point can support 80 pairs of comb capacitor plates with a capacitance of 1 pF. The capacitor plates are shown in Figure 1.

The designed accelerometer has four and a half pairs of comb structures as detection capacitors on one side. The total capacitance is 4.5 pF, the size of the device is 2.4 × 3.3 mm^2^, the size of the mass block is 0.6 mg, the resonant frequency of the device is 1.14 kHz, and the theoretical mechanical noise is 1.43 μg/√Hz. The displacement of the mass block caused by the unit gravitational acceleration is the capacitance sensitivity of 0.19 μm/g, and the theoretical capacitance sensitivity is 0.71 pF/g, all of which meet the design requirements.

The anti-overload design in the detection direction of structure was realized by distributed stopper structure. According to Yoon’s conclusions [8], when the stopper collides with the structure, if a flexible collision can be achieved, the stress caused by the collision will be greatly reduced, and the overload resistance of the structure will be improved. In addition to the low resonant frequency of the structure itself, the resonant frequency of an electrode plate is about 100 kHz, which is a good choice as a stopper [9]. Therefore, in this paper, a hypotenuse was designed between the mass block and the electrode plate to reduce the distance between them and the collision distance was reduced, as shown in Figure 2.

The stopper has two other advantages. First, the collision between the capacitor plate and the hypotenuse is an edge contact, which does not easily cause adhesion; second, although the collision distance between the hypotenuse and the electrode plate is reduced, the stopper is located in an open area and the etching time will not be increased.

## 3. Fabrication Process

The designed inclination accelerometer was prepared by the TGV process. The metal filling in the TGV was realized by the electroforming method, the structure layer was prepared by 110 μm-thick (100) crystal silicon, and the resistivity is less than 0.005 Ω cm. 

First, the AZ6130 photoresist was patterned on 110 μm-thick monocrystalline silicon, 10 μm-deep silicon was etched, and the bonding area was left. Magnetron-sputtered ITO served as the sacrificial layer, and stripping and cleaning were carried out. Then, the silicon wafer was bonded to a 300 μm-thick BF33 glass.

ITO also served as the mask for structural etching [10]. When ITO is used as a sacrificial layer, the material is transmits light, and the etching of the bottom of the silicon can be observed from the glass layer, avoiding the occurrence of over-etching. In addition, ITO materials with good conductivity can absorb the charged particles of the etching gas, reducing the footprint phenomenon caused by the reflection of F+ particles.

The release of the structure was carried out in sulfuric acid–hydrogen peroxide. After rinsing with pure water, it was boiled in MOS-grade alcohol for 30 min to displace the water in the structure, and dried on a hot plate. The preparation process is shown in Figure 3.

The advantage of ITO as an etching mask and sacrificial layer material is that the material has different corrosion resistance properties to silicon and glass. When etched in an acid solution, it can be effectively removed without affecting the structure. Compared with the traditional silicon oxide mask, the removal method is more economical and convenient. Using a photoresist as the mask, it is difficult to achieve high aspect ratio pattern etching on the bonding wafer with poor heat dissipation, and during etching, it is also difficult to ensure the etching accuracy of the structure due to the shrinkage of the edge of the photoresist.

The fabricated structures are shown in Figure 4. The designed plate distances are 4 μm and 8 μm, the etched sizes are 3.959 μm and 8.125 μm on the top side of the silicon, and the sizes are 4.306 μm and 8.472 μm on the backside. These values show excellent agreement between design and actual values, and the footprint is effectively suppressed.

The TGV cover plate needed to be prepared. First, laser drilling was performed on 300 μm-thick glass [11], the via was designed to the shape of a square, the length of the inlet side was 120 μm, and the length of the outlet side was about 90 μm. The metal filling in the hole was realized by metal casting mold [12], and the excess metal was removed through the CMP process. After CMP, CrAu was used as a mask for glass etching, and the cover plate was prepared. The process is shown in Figure 5.

Anodic bonding was used to realize the packaging of the tilt accelerometer. When bonding, a three-layer bonding method was used. A positive voltage of 280 V was applied to the silicon structure, and the glass substrate and the TGV cover were grounded, as shown in Figure 6. An excessively high bonding voltage was likely to cause the structure to be adsorbed upward or downward on the glass cover. To ensure bond quality at low voltages, good plasma cleaning of the structure and cover was required. Finally, the interconnection between different electrode groups was achieved through metal traces.

The preparation method used silicon and glass as the main materials, and the cost was low. In addition, the anodic bonding method was used to realize the wafer-level packaging, which also has good air tightness. The oxygen released in the anodic bonding increased the mechanical damping. The anti-vibration capability of the structure was improved. The processed device is shown in Figure 7.

## 4. Experiment Results

The dynamic signal analyzer HP35665A was used to test the frequency response characteristics of the accelerometer. It was found that the position of the frequency response curve fluctuated with the bias voltage, but no obvious harmonic response characteristics were measured. So, the released gas in the packaging process provided sufficient damping.

A CV test was carried out on the device using Keysight’s B1500, and the test results are shown in Figure 8. The structure of the device on the test surface had a good response to the electrostatic force, the etching and release process of the device was good, and the device was intact and free of adhesion. The capacitances between the mass and the positive and negative electrodes were about 5.52 pF and 5.32 pF, respectively. Considering the parasitic effect, the design value was consistent with the measured value. The differential capacitance deviation caused by the process was only 3.7%.

The capacitance detection was realized through a dedicated ASIC with digital output. The sensitivity of the accelerometer was tested on a centrifuge. The test results showed that the prepared accelerometer had good linearity in the range of plus or minus 2 g. The sensitivity of the accelerometer was 35,459.4 bits/g, and the corresponding sensitivity was about 28.2 μg/bit, which had high detection accuracy, as shown in Figure 9.

The accelerometer output was continuously sampled at 120 Hz for more than two hours when the input was zero, and the result is shown in Figure 10. The corresponding RMS noise was about 163.8 μg, the corresponding tilt noise was 0.0094°, and the noise density near 10 Hz was about 1.2 μg/√Hz, the test results are shown in Figure 10.

The Allan deviation of the accelerometer was calculated, and its zero-bias instability was 9.4 μg, which showed that the device has excellent performance as an inclination detection accelerometer, as shown Figure 11.

The device was bonded to the ceramic tube shell with DOWSIL™ ME-1030 die-bonding adhesive. The impact test was carried out 6 consecutive times at 10,000 g @ 0.5 ms in the detection direction. The structures of the tested 10 devices were not damaged after the impact, and the device function was good after the power-on test, indicating that the prepared inclination accelerometer had an anti-overload capability of 10,000 g.

## 5. Discussion

The accelerometer designed in this paper has low mechanical noise due to the use of 100 μm-thick silicon for the structural layer. The method of using ITO as the sacrificial layer reduces the foot effect and ensures the consistency of the device structure and the design value. Glasses with metal-filled through holes are used as the package cover, and the oxygen released by anodic bonding provides sufficient gas damping. Therefore, the accelerometer prepared by this process has extremely low mechanical noise and is suitable for high-precision inclination detection.

Compared with the devices that have been successfully commercialized on the market, the performance of the accelerometer prepared here also shows certain advantages, as shown in Table 1. Colibris’s and Murata’s pendulum accelerometers have larger masses and thus have lower mechanical noise. The advantage of Colibris‘s device is that the device is made of all-silicon material, including the upper and lower cover plates and the sensitive structure in the middle. When the ambient temperature changes, the deformation caused by thermal expansion is consistent, so it has excellent temperature stability and repeatability, which is incomparable for other accelerometers, but the cost is too high to be suitable for mass adoption in the industry. Murata’s upper and lower cover plates are made of silicon glass composite structure, and the gas-filled method is used to provide over-damping for sensitive structures during bonding, which has excellent reliability and good resistance to vibration and shock. However, the electrodes are grown on the sidewalls and cannot be screened at the wafer level, and the price is much higher than that of consumer-grade MEMS accelerometers.

Bosch has designed multi-layer materials as structural layers to achieve the purpose of increasing the size of sensitive structural mass blocks and reducing the impact of packaging stress on device performance, and is widely used in automobiles. The structure layer should be prepared by the epitaxial polysilicon method, and the thickness of the structure layer is about 30 μm. Compared with the sensitive structure prepared with 100 μm-thick single crystal silicon proposed in this paper, the noise control is slightly insufficient.

In Silicon Designs Inc., the low-noise accelerometer is realized by the double-seesaw process, which has lower noise, but the device is not packaged at the wafer level, which is very unfavorable for wafer-level testing and cost control.

The hinge-shaped accelerometer designed by Galtech has excellent sensitivity, and if the mass size is appropriately increased, then it can also achieve lower mechanical noise; however, this does not solve the packaging problem.

Although the sensitive structure of the high-performance MEMS accelerometer proposed in this paper adopts the traditional comb structure, the mechanical noise of the device is greatly reduced by increasing the thickness of the sensitive structure. Wafer-level packaging in glass with metal-filled through holes also has low-cost mass production capabilities and is a competitive implementation of high-performance MEMS accelerometers.

## Figures and Tables

**Figure 1 micromachines-13-01071-f001:**
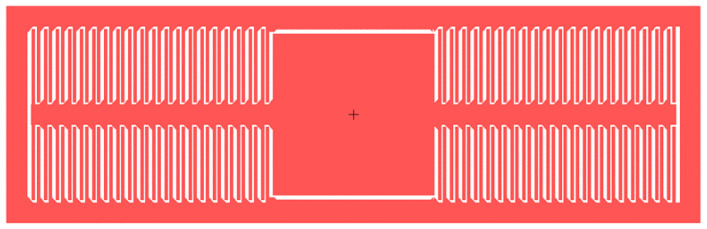
The comb capacitor.

**Figure 2 micromachines-13-01071-f002:**
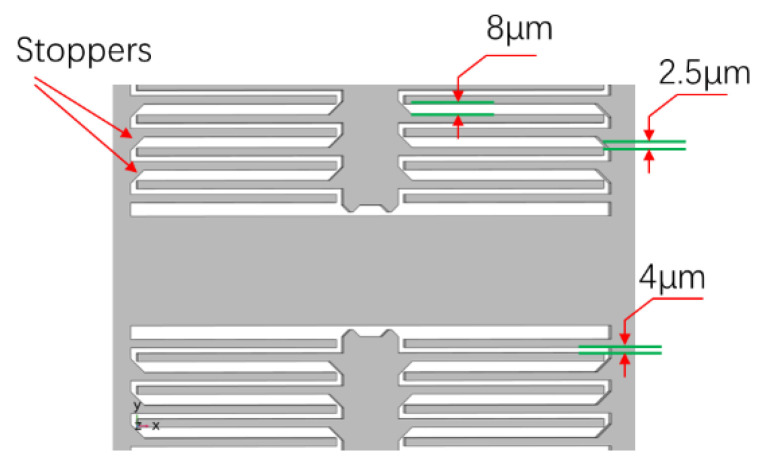
Stoppers located on electrodes.

**Figure 3 micromachines-13-01071-f003:**
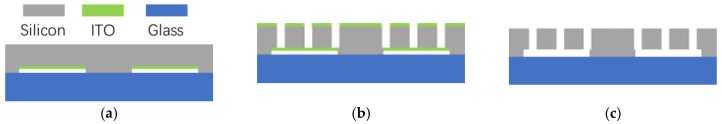
Preparation of the accelerometer structure: (**a**) preparation of structures; (**b**) structures etching; (**c**) structures release.

**Figure 4 micromachines-13-01071-f004:**
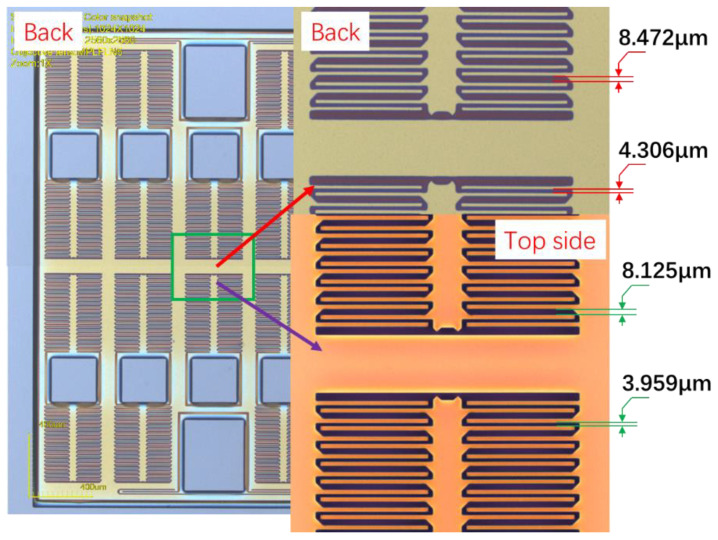
The etched structure.

**Figure 5 micromachines-13-01071-f005:**
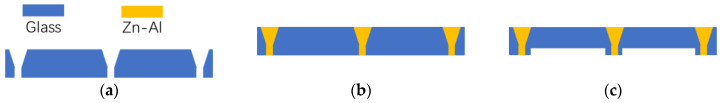
Preparation of the TGV cover: (**a**) laser drilling; (**b**) metal filling and CMP; (**c**) active area corrosion.

**Figure 6 micromachines-13-01071-f006:**
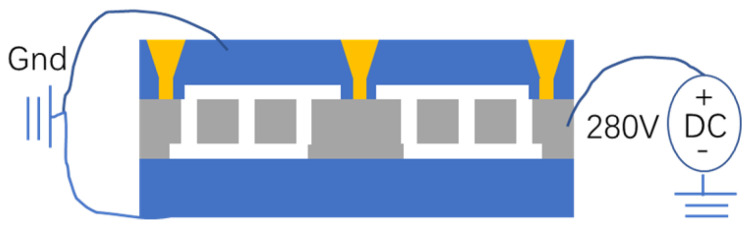
Anodic bonding was used for wafer-level package.

**Figure 7 micromachines-13-01071-f007:**
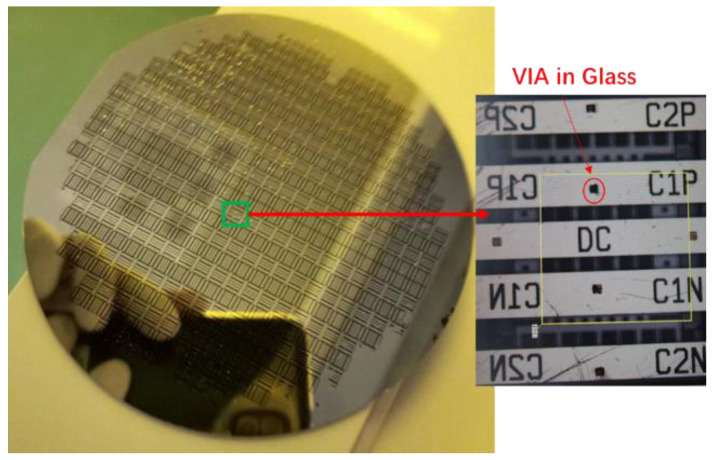
The processed accelerometers.

**Figure 8 micromachines-13-01071-f008:**
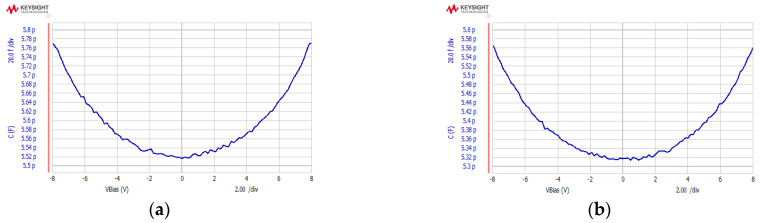
CV characteristics of the accelerometer: (**a**) positive direction; (**b**) negative direction.

**Figure 9 micromachines-13-01071-f009:**
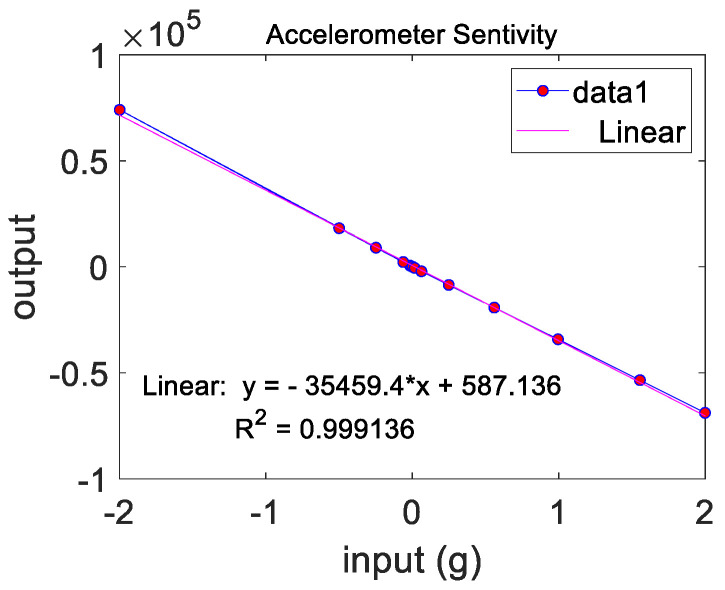
The test result of the scale factor.

**Figure 10 micromachines-13-01071-f010:**
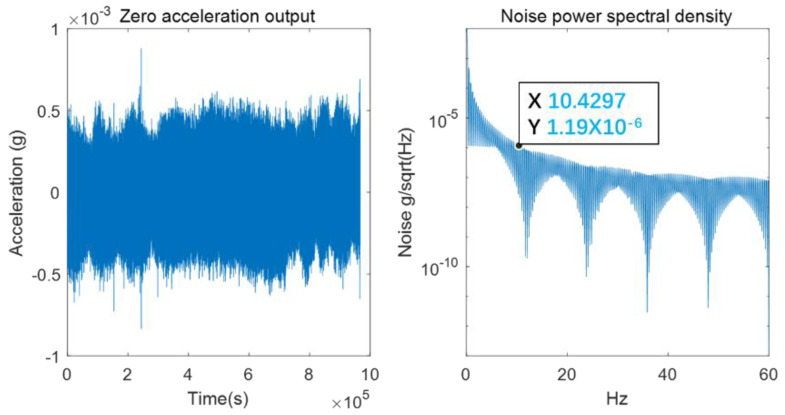
The zero-acceleration output.

**Figure 11 micromachines-13-01071-f011:**
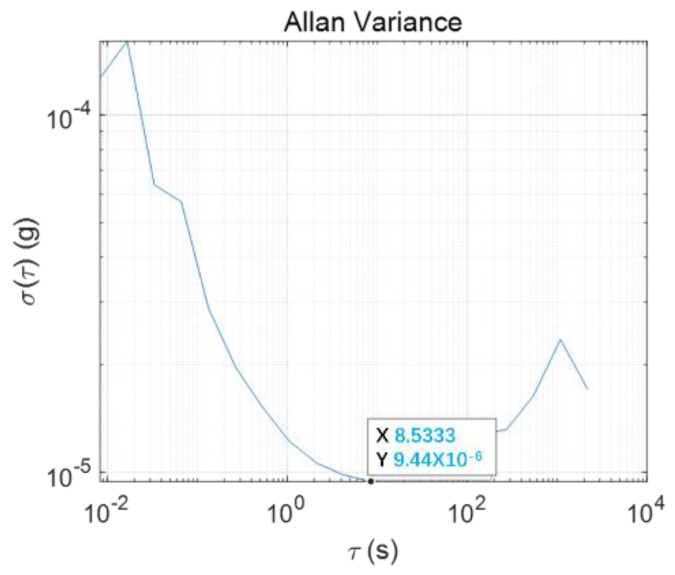
The Allan deviation.

**Table 1 micromachines-13-01071-t001:** Comparison of parameters about MEMS accelerometers for inclination detection.

Manufacturer	Model	Structure	Package Form	Noise (μg /√Hz)	Bandwidth (Hz)
Colibris [13]	RS9000	Pendulous	Silicon–Silicon	30	30–80
Murata [14]	SCA820	Pendulous	Anodic	220	18
ST [15]	IIS2ICLX	comb	Al-Si	15	12.5
BOSCH [16]	BMA456	Dual-layer comb	Al-Ge	120	1.5–1.6 k
SDI [17]	Model 1521	Teeter–totter		7	0–250
Gatech [18]		Hinge-shaped		72	
This Work		comb	Anodic	1.2	10

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
