# Peer review of "A High-Performance MEMS Accelerometer with an Improved TGV Process of Low Cost"

_micromachines, 2022, doi:10.3390/mi13071071_

Round 1

Reviewer 1 Report

This paper reports on the MEMS accelerometer.

Overall, although the paper provides measurement results of the fabricated accelerometer device, novelty of the paper is very limited. Among several things the authors mentioned in the paper (casting mold?, TGV?, distributed stopper design?, etc.), it is difficult to know which is the major improvement compared to the previous works. To my knowledge, most of those factors have already been demonstrated. The authors must clarify what is the originality and main contribution of this work.

- The title of this paper is somewhat vague and too broad to my feelings. It must be more specific, focusing on the major contribution of this work.

- Fig. 3 & 5: name of each part (layer) needs to be displayed.

- Fig. 4: cannot read the texts in the figure. Resolution of the figure must be improved.

- Section 4 (page 5, line 166~169): these claims need to be supported by experimental data.

- I strongly recommend you to put the comparison table with other works, focusing on the differences in terms of the performance and methodologies you used.

Reviewer 2 Report

Nice engineering work with good design methodolgy and well executed experimental work, but lacks novelty and the performance of the accelerometer prototypes are not compared with similar research work or products.

In introduction, you are writing about commercial comparable products and describes their performances without giving references nor product names. Performances are also not quantified. Either give this information or skip this text.

Introduction should address more similarresarch work and list them in the references.

Design and simulation, fabrication process and experimental work are adequately described. A recommendation is to put discussions in a separate chapter to let the reader objectively evaluate the experimental results.

Author Response

Thanks a lot for your suggestions

1 Performance description about commercial comparable products in the introduction part is insufficient, Thanks for pointing out.

I Think the process schemes of each company are different, and the influence on performance and cost is properly valued in the introduction regarding different models of products, so there were no detailed parameters in the introduction part, but the additional information is added in the fifth part (5. Discussions).

2 Table 1 is added to compare the differences between accelerometers with similar functions, and the evaluation of the experimental results is added.

  1. References to similar works are added in the 5th part.

Thanks a lot.

Round 2

Reviewer 1 Report

The authors revised the manuscript based on the comments.